# Analysis of Simian Endogenous Retrovirus (SERV) Full-Length Proviruses in Old World Monkey Genomes

**DOI:** 10.3390/genes13010119

**Published:** 2022-01-10

**Authors:** Antoinette C. van der Kuyl

**Affiliations:** Laboratory of Experimental Virology, Department of Medical Microbiology and Infection Prevention, Amsterdam UMC, University of Amsterdam, Meibergdreef 9, 1105 AZ Amsterdam, The Netherlands; a.c.vanderkuyl@amsterdamumc.nl; Tel.: +31-205-666-778

**Keywords:** endogenous, retrovirus, SERV, Old World monkey, phylogeny, evolution, paleovirology

## Abstract

Simian endogenous retrovirus, SERV, is a successful germ line invader restricted to Old World monkey (OWM) species. (1) Background: The availability of high-quality primate genomes warrants a study of the characteristics, evolution, and distribution of SERV proviruses. (2) Methods: Cercopithecinae OWM genomes from public databases were queried for the presence of full-length SERV proviruses. A dataset of 81 Cer-SERV genomes was generated and analyzed. (3) Results: Full-length Cer-SERV proviruses were mainly found in terrestrial OWM, and less so in arboreal, forest- dwelling monkeys. Phylogenetic analysis confirmed the existence of two genotypes, Cer-SERV-1 and Cer-SERV-2, with Cer-SERV-1 showing evidence of recent germ-line expansions. Long Terminal Repeat (LTR) variation indicated that most proviruses were of a similar age and were estimated to be between <0.3 and 10 million years old. Integrations shared between species were relatively rare. Sequence analysis further showed extensive CpG methylation-associated mutations, variable Primer Binding Site (PBS) use with Cer-SERV-1 using PBS^lys3^ and Cer-SERV-2 using PBS^lys1,2^, and the recent gain of LTR motifs for transcription factors active during embryogenesis in Cer-SERV-1. (4) Conclusions: sequence analysis of 81 SERV proviruses from Cercopithecinae OWM genomes provides evidence for the adaptation of this retrovirus to germ line reproduction.

## 1. Introduction

Mobile genetic elements, including viruses, abound in living organisms. They add considerably to the flexibility and gene content of both pro- and eukaryotic genomes. After deleterious insertions from the germ line were purged, many of the Endogenous Viral Elements (EVEs) left behind have mutated over time and some have been adopted by the host. Undeniably, such processes are of great value when studying the reshaping of genomes, but an often forgotten aspect of EVEs is the information that captured viral sequences can give us about the deep history of the viruses themselves. By now it has been firmly recognized that EVEs are derived from viruses that were once infectious; in fact, endogenization of exogenous retroviruses can be seen in action in several species, for instance, murine leukemia virus (MLV) in mice, feline leukemia virus in cats, and koala retrovirus in koalas (for an overview, see [1]). Likewise, many viruses, and especially retroviruses, where chromosomal integration is an essential part of their lifecycle, have entered the mammalian germ line in the past, with some being more successful in doing so than others (for a review, see [2]). An example of a successful colonizer in relatively recent times, with germ line activity dated to around <0.3–8 million years ago (mya), is the Old World monkey (OWM) type D betaretrovirus simian endogenous retrovirus, SERV [3,4]. A related virus, simian D type retrovirus (SRV), is at present infectious and pathogenic in macaques [5].

Full-length or nearly full-length SERV proviral genomes are found in many OWM species, with up to 80, mostly heterozygous, integrations detected in individual monkeys [6]. A variant SERV, col-SERV, with full-length integrations and substantial homology to SRV-6 (simian type D retrovirus-6), was reported from the Asian Colobine species *Rhinopithecus roxellana* and *Rhinopithecus bieti*, with incomplete proviruses being present in *Nasalis larvatus* [7]. Phylogenetic analysis has suggested that Col-SERV had spread between and integrated in the genome of Colobine ancestors after the separation of Cercopithecinae and Colobinae, which has been estimated to have occurred approximately 16 mya [7,8]. Although the genomic features and distribution of SERV in Cercopithecinae OWM species have been published [3,4], the substantial number of complete SERV genomes present in Cercopithecinae OWM genomes now enables a comparison at the population level. Viral variation before integration as well as mutation patterns due to host activity after integration can be analyzed to learn more about the population structure of the once infectious virus as well as about host repression mechanisms acting upon the parasitic elements. However, a restriction is that it is unlikely that all ancient viral variation of the past epidemic has been captured and preserved in extant primate germ lines.

Analyses show that Cercopithecinae SERV, Cer-SERV, was a highly variable virus, with two major genotypes that differ in their primer binding site (PBS) use. PBS^lys1,2^ sequences, present in Cer-SERV-2 genomes, suggested repeated escape from tRNA fragment (tRF) repression, a mechanism operational in stem cells [9]. Such escape signals were not apparent in the PBS^lys3^ of Cer-SERV-1 integrations. Furthermore, mutation patterns suggest that CpG methylation of the long-terminal-repeat (LTR) region has been a major repressive mechanism to restrict viral replication. In contrast, APOBEC-induced mutation patterns were rarely observed. Cer-SERV-2 integrations appear to be somewhat older than those of Cer-SERV-1. A low number of shared integrations was seen between baboon and gelada (Cer-SERV-2) and between macaque species (Cer-SERV-1 and -2). Younger expansions of full-length Cer-SERV-1 proviruses are present in some OWM species, where Cer-SERV-2 appears to be less active with respect to reinfections of the germ line.

## 2. Materials and Methods

### 2.1. Identifying Simian Endogenous Retrovirus (SERV) Sequences in Cercopithecinae

SERV 23.1 (GenBank acc. no. U85505; [3]), a full-length provirus with a genome length of 8393 nt, obtained from a *Papio cynocephalus* (yellow baboon) chromosomal DNA library, was used to identify endogenous SERV nucleotide sequences in OWM genomes from GenBank (www.ncbi.nlm.nih.gov/genome/ (accessed on 12 October 2021)) and Ensembl (www.ensembl.org/index.html (accessed on 12 October 2021)) databases by similarity search using the BLASTn/BLAT algorithm provided.

Default settings were used in the NCBI BLASTn search. As the aim of the research was to retrieve and describe specifically SERV integrations and not more distantly related endogenous viruses, searches were optimized for “highly similar sequences” (megablast). Only when no results were retrieved with megablast was the discontiguous megablast option (“more dissimilar sequences”) used. In Ensembl BLAST searches, search sensitivity was the default (“normal”), and filtering of low complexity regions or filtering of query sequences with RepeatMasker was disabled. As SERV 23.1 is a SERV genotype 1 provirus, additional searches were done with a SERV-2 provirus retrieved from a *Papio anubis* genome.

Cercopithecinae primate genomes queried were *Cercocebus atys, Cercopithecus mona, Cercopithecus neglectus, Chlorocebus sabaeus (*formerly known as *Cercopithecus aethiops sabaeus*)*, Erythrocebus patas, Macaca fascicularis, Macaca fuscata, Macaca mulatta, Macaca nemestrina, Mandrillus leucophaeus, Mandrillus sphinx, Papio anubis* (monkey no. 1X1155 and no. 15944), and *Theropithecus gelada*. For *C. sabaeus*, a genomic assembly from a male monkey as well as genomes from the Vero cell line, derived from a female, are available. For the olive baboon, *P. anubis*, there are also two genomes in the database: Panubis1.0 (male monkey no. 15944) has been reported to be more accurate than the earlier Panu_3.0 (female monkey no. 1X1155) assembly [10,11].

### 2.2. Analysis of Proviral Sequences

BLAST results were downloaded from the databases, and full-length or nearly full-length proviral genomes (covering > 95% of the query sequence) were selected. Sequences were aligned using ClustalW as implemented in BioEdit (bioedit.software.informer.com (accessed on 5 November 2021)); alignments were optimized through visual inspection. In order to include only genuine SERV proviruses, viral genomes whose 5′ and 3′ Long Terminal Repeat (LTR) did not cluster at all were considered to be the result of an assembly error and were deleted from the alignment. Low-quality sequences, which included a full-length provirus from *M. leucophaeus* (drill) and sequences from *M. nemestrina*, were likewise removed from the dataset. A 6 nt duplication flanking the provirus was taken as evidence of virus-driven integration. The alignment is available in Appendix A.

Nt distances were calculated with the Kimura-2-parameter method and the evolutionary history was inferred using the Maximum Likelihood method and Hasegawa-Kishino-Yano model [12]. Initial tree(s) for the heuristic search were obtained automatically by applying Neighbor-Join and BioNJ algorithms to a matrix of pairwise distances estimated using the Maximum Composite Likelihood (MCL) approach, and then selecting the topology with superior log likelihood value. A discrete γ distribution was used to model evolutionary rate differences among sites (five categories (+G, parameter = 0.3608)). All positions with less than 95% site coverage were eliminated; i.e., fewer than 5% alignment gaps, missing data, and ambiguous bases were allowed at any position (partial deletion option). Evolutionary analyses were conducted in MEGA X [13]. The phylogenetic tree was visualized using FigTree version 1.4.3. Recombination between *T. gelada* Cer-SERV-1 and -2 genomes was investigated using RIP3.0 [14]. Tajima’s Neutrality test, as implemented in MEGA X, was done for all four aligned Cer-SERV-1 and Cer-SERV-2 ORFs, excluding those with out-of-frame insertions or deletions (indels).

The formula T = K/2r, whereby K is the pairwise nt distance between the 5′ and 3′ LTR sequences and r is the substitution rate, was used to calculate the age of the full-length proviruses.

APOBEC-induced hypermutation was analyzed with the HYPERMUT 2.0 tool available at hiv.lanl.gov/HYPERMUT/hypermut.html [15]. Pattern definitions were adjusted to detect either APOBEC3G or -non-3G patterns and to exclude CpG sites. Thus, settings were K-(G→A)-D for APOBEC3G mutation patterns with K-(G→A)-N for control sites, and D-(G→A)-AD for non-3G mutations, with D-(G→A)-N for corresponding control sites. The context was enforced on both reference and subject sequences. HYPERMUT is sensitive to the alignment and to the choice of reference sequence, so that results obtained give only an approximation of potential mutation patterns [16]. Fisher’s exact test, as implemented in the program, was used to assess statistical significance. For CpG-site associated mutations (TpG and CpA), HYPERMUT settings were: C→T-G and C-G→A for each mutation, respectively, and C→C-G for the controls.

Transcription factor binding sites (TFBS) in the SERV 5′ LTR were predicted using the Match and P-Match options with the TRANSFAC database at gene-regulation.com (accessed on 7 September 2021). Settings included the use of only high-quality vertebrate matrices, with a cut-off selection for the matrix to minimize the sum of both false positives and false negatives.

## 3. Results

Sequences homologous to Cer-SERV were retrieved from all analyzed OWM genomes, but full-length proviral insertions with >90% homology to the query sequence were restricted to *C. sabaeus*, *E. patas*, *M. leucophaeus, P. anubis*, *T. gelada*, and the Asian macaque species *M. fascicularis*, *M. fuscata*, *M. mulatta*, and *M. nemestrina* (Table 1).

Using sequences deposited in genome databases has a few caveats:Sequence and assembly quality may vary;Not all species have been sequenced;For most species, only a single genome assembly is available;Y chromosome sequences are not always present;Heterozygous EVEs are omitted from assemblies;Full-length proviruses may be younger than fragmented ones;It is not likely that all viral variation has been captured in the germ line.

A total of 79 full-length, good-quality SERV genomes were retrieved from the monkey genomes listed above. Fifty-three genomes clustered with the Cer-SERV-1 genotype, and 26 with the Cer-SERV-2 genotype [3]. The majority, namely 26 SERV proviruses, were found in the gelada genome. The numbers reflect previous research, which had already indicated that SERV is the most successful invader of the OWM germ line and that its copy numbers greatly exceed those of other monkey endogenous retroviruses [4].

Therefore, a dataset of 81 reliable SERV genomes, 79 novel ones retrieved during this study, the original SERV 23.1 proviral integration, as well as a full-length SERV genome isolated from the Vero cell line (GenBank acc. no. AB935214 [17]), is used in the current study. Both reference sequences are genotype 1, so that in total 55 Cer-SERV-1 genomes are available for analysis.

### 3.1. Phylogenetic Analysis of Cercopithecinae SERV Proviruses

#### 3.1.1. SERV Diversity in OWM

A phylogenetic analysis of the OWM full-length Cer-SERV genomes shows that clustering of the sequences in two genotypes, Cer-SERV-1 and Cer-SERV-2, described in an earlier study of a SERV pol-env fragment [3], is confirmed when using longer sequences (Figure 1). The two genotypes have an approx. 88% nucleotide homology, and both are equidistant to Col-SERV proviruses from *Colobus angolensis*, with around 72% nucleotide similarity in a 90% query cover. 

In general, Cer-SERV sequences cluster randomly with respect to their host species, but species-specific clusters can be observed (Figure 1). The topology of the tree has likely been influenced by ancient interspecies transmissions, incomplete lineage sorting, and recent activity including reinfection of the germ line and/or genome duplications. Shared proviruses, as determined by integration site analysis after visual inspection of the tree, are rare but do exist, for instance, between macaque species and between *Papio* and *Theropithecus*.

The nucleotide composition of the SERV genome, A-rich and G-poor, is in line with that reported for primate type D retroviruses (whereby A > U > C > G), and recapitulates the recombinant origin of the virus group, which carry type B gag-pol genes (A > U > G > U) and a type C env ORF (C > A > G > U) (Appendix A) [18,19]. The A-nucleotide percentage is decreased in the env ORF (~29.7% A) compared with the gag-pol coding region (~33.3% A). A low A-count, ~22.6%, is found in the LTR, a common finding for promoter sequences, which, in vertebrates, are GC-rich [20].

The majority of mutations in the dataset are transitions (ML estimate of transition/transversion bias R = 4.44 for Cer-SERV-1 and 3.56 for Cer-SERV-2), with G to A and C to T changes being the most frequent (Appendix A). These transitions mostly occur at CpG dinucleotides. T to C mutation levels are also relatively high, mainly because CpG dinucleotides have, in a majority of sequences in the alignment, been mutated to TpG so that the putative ancestral C is now perceived as the mutation. Redrawing the phylogenetic tree without CpG sites in the alignment did shorten the branch lengths, but not the topology of the tree, nor the occurrence of the specific clusters, suggesting that most of the nucleotide variation in SERV genomes was already present before integration. 

#### 3.1.2. Cer-SERV Integrations in the *T. gelada* Genome

In the current study, a considerable number, namely 26, of the Cer-SERV full-length proviruses were retrieved from the gelada genome. Such a result does not imply that, among OWM, geladas possess the highest SERV copy number, but rather that many proviruses in the genome are present in a homozygous state. The limited geographical distribution of extant geladas, possibly leading to inbreeding, which facilitates the fixation of traits, could likely explain that observation. As such, a substantial number of sequences can give a better understanding of SERV evolution, the characteristics of *T. gelada* SERV will be described in more detail. Of the 26 proviruses, 21 are genotype Cer-SERV-1 and 5 are genotype Cer-SERV-2. Approximately half (46%) of the proviruses are found on the plus strand, suggesting little selection bias with respect to genomic orientation. However, four proviruses that overlap with protein-coding genes are all in the reverse orientation, suggestive of a negative selection of forward orientated integrations, possibly because of the introduction of RNA polymerase II transcription termination sites (TTS) provided by the LTR [21]. Such transposable element-associated TTSs are known to cause premature, aberrant termination of host gene transcription. A provirus integrated next to a 5S rRNA gene exon on chromosome 19 is in the forward orientation again, which is probably less of a problem here, since tRNA genes are transcribed by RNA polymerase III. None of the full-length genomes show any evidence of recombination between Cer-SERV-1 and Cer-SERV-2, suggesting that the lineages were probably not actively replicating at the same time or did not infect the same cell.

Besides the full-length genomes, at least 150 partial proviruses, ranging from ±500–8280 nt in length, are found in the gelada genome, with integrations being present on every chromosome. Unfortunately, the Y chromosome could not be assessed, as the available genome has been derived from a female gelada. Phylogenetic analysis showed that the variation present in the partial genomes is adequately represented by the full-length genomes. Partial genomes cluster among the full-length proviruses and have similar branch lengths, suggesting that there is no significant difference in age. The majority of defective proviruses represent internal fragments of the coding regions, lacking most or all of the LTRs. Such randomly defective proviruses could possibly have been generated during chromosomal recombination, or a comparable process, in the fertilized oocyte. In addition, around 45 solo LTRs, left behind when a provirus is excised after homologous recombination between the 5′ and 3′ LTR [22], are found in this gelada assembly. Seven of these solo LTRs are present within protein-coding genes, three of them in the forward orientation.

### 3.2. Mutation Patterns in SERV Proviruses

#### 3.2.1. CpG Methylation-Associated Mutations

The mutation pattern in the SERV dataset is dominated by transitions, especially G to A and C to T, with a transition/transversion bias R of 4.23, as estimated by Maximum-Likelihood analysis. The majority of G/A and C/T transitions are part of CpG dinucleotides, suggesting that deamination of 5-methylcytosine to thymine after CpG methylation led to the observed variation. Host-induced cytosine methylation of CpG dinucleotide sequences by DNA methyltransferases is a mechanism to repress RNA expression and is commonly used to silence proviral integrations. CpG sites have the fastest mutation rate observed in vertebrate genomes, estimated at 1.2–2.9 × 10^−8^ substitutions/site/year in primate genomes [23], whereby transitions at human CpG sites have been determined to occur at a rate of 1.6 × 10^−7^ s/s/y [24].

Analyzing the CpG mutation pattern of SERV genomes in more detail reveals that, in the gelada dataset, approximately 6% of the on average 198 CpG sites/genome have been mutated in Cer-SERV-1, while for Cer-SERV-2, approximately 15% of the on average 144 CpG sites have been mutated. In each case, about half of the mutations per genome concern substitutions to TpG and the other half to CpA for both genotypes (result not shown), implying that CpG methylation is not orientation-dependent in these proviruses. Such a pattern is also observed in Cer-SERV genomes from other OWM species, but as the recognition of CpG sites is dependent on the sequence alignment, mutations are less easy to quantify here, especially when gaps are present (results not shown). Overall, the analyses confirm that CpG-methylation-associated mutations are highly prevalent in Cer-SERV, and that in Cer-SERV-2 proviruses, a higher percentage of CpG sites has been mutated, suggestive of a longer residence in the host genome and thus of an older age. In the Cer-SERV-1 cluster, there are basal sequences with long branches in the phylogenetic tree and a cluster with much shorter branches (Figure 1). In *T. gelada* Cer-SERV-1, the percentage of CpG mutations in the long-branch cluster is 10.5% versus 4.7% in the short-branch cluster, implying age variation between the SERV-1 proviruses.

#### 3.2.2. APOBEC3G and -Non-3G Signature Patterns

The apolipoprotein B mRNA editing enzyme catalytic polypeptide-like (APOBEC) subfamily 3 members are powerful virus restriction factors. They are able to edit retroviral minus-strand DNA genomes during replication, thus before integration, possibly through cytidine deamination, resulting in G to A mutations in the plus-strand leading to sometimes “lethally” hypermutated genomes (see [16]). Such hypermutated genomes can be integrated, and are found in human immunodeficiency virus (HIV) infected individuals [25]. Many retroviruses are resistant to certain APOBEC3 family members, by for instance avoiding packaging APOBEC3G proteins in the viral particle (MLV and Mason-Pfizer monkey virus (MPMV)) or by expressing anti-APOBEC3 viral factors (such as Vif and Bet, encoded by HIV and foamy viruses, respectively, which bind APOBEC3G) [26,27,28]. APOBEC3 proteins can also inhibit LTR-transposon activity, and a resurrected HERV-K(HML-2) endogenous betaretrovirus was shown to be susceptible to APOBEC3F, but not to −3G [29]. Nonetheless, two HERV-K(HML-2) EVEs did show evidence of extensive APOBEC3G hypermutation [30,31]. In primates, the APOBEC3 family has recently expanded and is under strong positive selection [32,33]. It is, however, unknown whether or not primate D-type retroviruses can resist APOBEC3 proteins; the related betaretrovirus MPMV is resistant against APOBEC3G [27].

When analyzing the SERV dataset for APOBEC3G and -3F mutations, it was first observed that there are no lethally hypermutated genomes among the sequences, although a single provirus on *T. gelada* chr. 7b showed some evidence of hypermutation (Fisher’s exact test, *p* = 0.005 for the APOBEC-non-3G pattern). Hypermutated sequences were not present in the partial virus genome dataset from *T. gelada*. GG to GA mutations, observed after APOBEC3G editing, are relatively rare in SERV genomes. GA to AA mutations, characteristic for non-3G APOBEC3s such as APOBEC3F, can be found in slightly higher numbers in the alignment, but all *p*-values were non-significant for comparisons (result not shown). Rate ratios, which give an indication of the number of observed mutations with regard to the number of possible mutations, are on average <1 for APOBEC3G patterns, and ≥1 for APOBEC-non-3G mutations, suggesting some action of APOBEC-non-3G proteins but not APOBEC3G. Overall, these results indicate that SERV would be resistant to APOBEC3G, similar to MPMV, but could be susceptible to APOBEC3F editing.

#### 3.2.3. PBS Variation

One of the most remarkable differences between the two Cer-SERV genotypes is the primer binding site (PBS) sequence immediately downstream of the 5′ LTR, which binds the tRNA primer needed for first-strand DNA synthesis. The Cer-SERV-2 PBS is complementary to tRNA^lys1,2^ (5′ TGGCGCCCAACGTGGGGC 3′), while in Cer-SERV-1, it is in the majority of proviruses complementary to tRNA^lys3^ (5′ TGGCGCCCGAACAGGGAC 3′) except in two gelada SERV-1 integrations from *T. gelada* and in one from *Chlorocebus*, where the PBS is the wt PBS^lys1,2^, sequence (Figure 2). Seven closely related Panubis1.0 SERV-1 proviruses have a variant PBS^lys1,2^ sequence. These observations suggest that PBS switching is common in SERV. A combination of a Cer-SERV-1 LTR with a PBS^lys1,2^ sequence could not be found in Panu_3.0. Together with the near absence of shared integrations between the olive baboon genomes, the lack of such a virus variant does suggest that many proviruses in *P. anubis* are present in a heterozygous form, and are excluded from genome assemblies. A similar observation has been published regarding the extensive heterozygosity of SERV proviruses in the African green monkey genome [6]. Of the currently circulating primate retroviruses, SRV and simian T-cell leukemia virus (STLV) do contain a PBS^lys1,2^, while simian immunodeficiency virus (SIV) uses tRNA^lys3^ as a primer. Within these viral species, such variable use of tRNA^lys^ as seen in SERV has not been observed. On the other hand, copies of the murine VL30 retrotransposon family have been reported to harbor alternatively PBS^pro^, PBS^gly^, or PBS^gln^ [34].

As for variation within the PBS, C to T substitutions at the two CpG sites of the Cer-SERV-1 PBS^lys3^ are seen in 9/46 genomes, in three at the first, and in six at the second CpG (Figure 2). Alternatively, the C8T mutation could point to the use of tRNA^lys5^, a gene variant of tRNA^lysUUU^, for priming. tRNA^lys5^ is used at low frequencies by human immunodeficiency virus, HIV [35,36], and by SIV [37]. Variation is also seen in the SERV-2 PBS^lys1,2^.

**Figure 2 genes-13-00119-f002:**
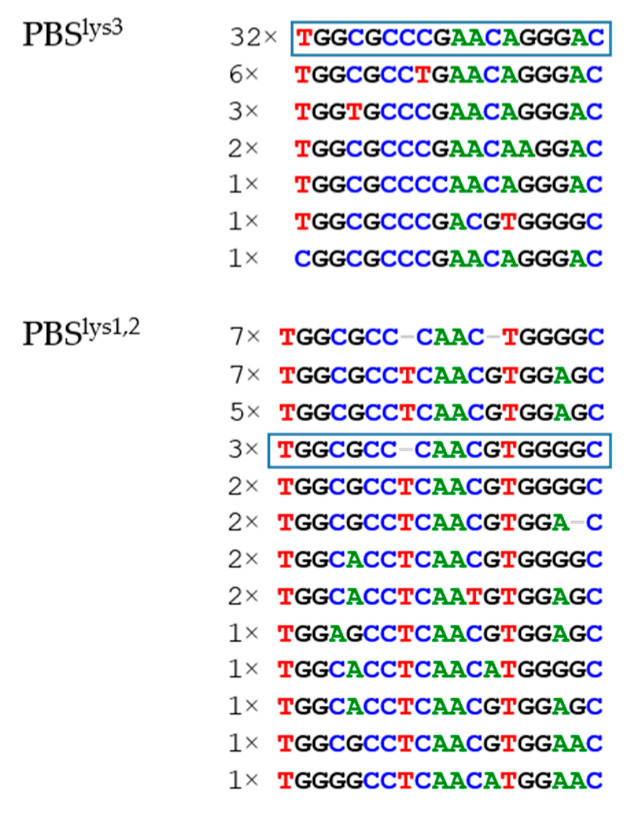
PBS sequences present in 81 full-length Cer-SERV genomes. The consensus sequence of PBS^lys3^ and PBS^lys1,2^ is boxed. The number of viral genomes with each variant sequence is indicated. Forty-six Cer-SERV-1 genomes harbor a PBS^lys3^, 9 SERV-1 genomes have a PBS^lys1,2^ or a sequence more closely related to PBS^lys1,2^. Twenty-six Cer-SERV-2 genomes have a PBS^lys1,2^, three of which are of the wild-type sequence; all seven PBS^lys1,2^ sequences without a T and a G nucleotide are from a Cer-SERV cluster in *P. anubis* 15944. Two of three wt PBS^lys1,2^ sequences are Cer-SERV-1 sequences from *T. gelada*, and the third is a Cer-SERV-1 provirus from *C. sabaeus*.

Where the majoritycontain an additional T nucleotide (Figure 2). In a murine retrovirus, 2–3 mutations in PBS^lys1,2^, including a single nt insertion, have been shown to be tolerated in replication while conveying escape from tRNA fragment (tRF) repression [38]. HIV-1 can likewise allow 1–2 nt insertions in its PBS [39]. Other variation observed in the Cer-SERV-2 PBS sequences are C to T or G to A mutations at CpG dinucleotides. Of note, in PBS^lys1,2^, all but one CpG associated mutation is a G to A substitution—in 6/26 proviruses at the first CpG, and in 2/23 at the second CpG—suggesting modification of the opposite strand compared with PBS^lys3^. In addition, GGGG at the 3′ end has often been modified to GGAG (16/26) or GGAA (2/26), suggestive of APOBEC3G and/or APOBEC3F action. Thus, mutation patterns of the PBS appear to vary according to the specific sequence, with PBS^lys1,2^ showing an increased mutation load compared with PBS^lys3^, which could be related to an older age, or to more effective repressive mechanisms for PBS^lys1,2^ or both. Adapting PBS sequences would then be an answer to such repression.

### 3.3. Genomic Features of SERV Proviruses

#### 3.3.1. Cer-SERV LTR Sequence Diversity and Dating

The LTR sequences of Cer-SERV-1 are on average 484 nt (range 439–495 nt) in length, with the shorter LTRs, having a deletion of either 23 or 42 nt around nt 220, being found in *C. sabaeus* proviruses. Using the TRANSFAC database, between 5 and 8 TFBSs were predicted on the positive strand of each 5′ LTR, among which detection of motifs for AREB6, Brn-2, CDP, CREB, E2F, FoxD3 (2 × /LTR), and FoxJ2 was common. The upstream FoxD3 motif was only found in a subset of younger SERV-1 genomes, suggesting it has been a relatively recent acquisition. The FoxJ2 motif, present in 42/51 Cer-SERV-1 5′ LTRs, was gained after a G to A mutation in a CpG motif. In *C. sabaeus* Cer-SERV-1, a second FoxD3 motif was found downstream of the first motif in 3/5 LTR sequences in a duplication next to a large deletion. Interestingly, the expression of FoxD3 is limited to embryonic stem cells, and FoxJ2 is highly expressed during spermatogenesis and in early embryonic development [40,41].

Cer-SERV-2 LTRs are on average 486 nt (range 475–486 nt) in length. Compared with SERV-1, TFBS analysis suggests a slightly different set of transcription factors to be able to bind to the 5′ LTR of Cer-SERV-2. Around 5–10 TFBS per LTR were predicted here, with those for AREB6, COMP1, CREB, NF-1, RFX1, Pax6, and v-Myb being detected repeatedly. FoxD3 and FoxJ2 motifs were not reported for Cer-SERV-2 LTRs.

LTR divergence in retroviruses is a useful measure of time elapsed since integration, whereas the 5′ and 3′ LTR are identical due to the retroviral replication strategy, after which they start to diverge with the host mutation rate [42,43]. Using LTR nt distance, SERV integration times in Cercopithecinae have been estimated at an average of 3.42 ± 2.20 mya and 6.16 ± 3.41 mya in Colobinae [7]. However, deciding which r to apply is complicated, and several rates have been proposed. Based on primate ERVs, r has been estimated to be in the range of 2.3–5 × 10^−9^ substitutions/site/year (s/s/y) [43], somewhat faster than the average mutation rate of 1–2.2 × 10^−9^ s/s/y calculated for vertebrate genomes. An important factor in determining r is the number of CpG methylation-related mutations, which abound in ERVs and are known to occur at an increased rate compared with other mutations [23,24]. An option could be to ignore CpG-associated mutations in calculations (see [44]), but such an approach would remove most of the variation for SERV.

The mean nt distance within the dataset was similar between the 5′ and 3′ LTRs for Cer-SERV-1 (5′ LTRs: 0.090, 3′ LTRs: 0.103) and Cer-SERV-2 (5′ LTRs: 0.064, 3′ LTRs: 0.069), suggesting a comparable mutational pressure. However, nt distances between LTRs from single proviruses showed considerable variation (Table 2 and Appendix A). Both the mean age of Cer-SERV-1 integrations and the integration times of individual proviruses are more recent than for Cer-SERV-2, although the calculated age ranges overlap. Estimated integration times of Cer-SERV proviruses in OWM using the more appropriate, faster mutation rate are shown in Figure 3. Using similar substitution rates, Ikeda et al. dated seven full-length Cer-SERVs in the *P. anubis* genome, irrespective of genotype, to range between T_int_ = 2.09–19.03 mya when assuming r = 2 × 10^−9^ s/s/y, and T_int_ = 0.84–7.61 mya when r = 5 × 10^−9^ s/s/y [7], which corresponds with the dating done here.

#### 3.3.2. Cer-SERV Coding Region Diversity

In SERV, the coding region encompasses the major retroviral gag, dUTPase, protease, pol, and env genes, but it does not contain open reading frames (ORFs) for any additional proteins, as seen in the lenti- and betaretrovirus families. Analysis of the coding capacity of the 81 Cer-SERV genomes suggests that many of them contain uninterrupted ORFs. In fact, in six Cer-SERV-1 and one Cer-SERV-2 provirus(es), all major genes are open (Appendix A). In the other genomes, genes are mainly interrupted by a single nt indel, commonly in a stretch of the same nucleotides. It is not implausible to assume that many of these indels originate from the sequencing approach, so that more intact ORFs may be present. In other genes, premature stop codons are found, which often arise from CpG methylation-associated mutations. Of course, non-interrupted genes are a requirement for protein expression but do not imply that the encoded proteins are functional, as deleterious amino acid mutations may be present. Furthermore, synonymous mutations in codons, which do not affect protein function, can influence RNA motifs, RNA structure, and translation and may thus not be neutral at all.

To examine possible selection on Cer-SERV genes, Tajima’s Neutrality test was performed using aligned sequence sets of the Cer-SERV ORFs. All D-values were strongly negative (range −0.47 till −1.67 for Cer-SERV-2 gag and protease, respectively), suggestive of a recent selective sweep or population expansion after a recent bottleneck.

It is of interest to note that, compared with gag (Cer-SERV-1: 55%; Cer-SERV-2: 23%) and pol (Cer-SERV-1: 71%; Cer-SERV-2: 39%), a lower percentage of Cer-SERV-1 proviruses has an uninterrupted env gene (Cer-SERV-1: 40%; Cer-SERV-2: 27%), which is significant with respect to pol for Cer-SERV-1, but not Cer-SERV-2 (Two-tailed Fisher’s exact test, *p*-value for comparison = 0.002). However, because of the short evolutionary time and the possibility of complementation, it may be too early to conclude that SERV is on the way to lose its env gene, which would no longer be needed should the host DNA be directly targeted by the integration complex. This implies we cannot yet distinguish between the two infection mechanisms. Although it is striking that the largest ORF, pol (2613 nt), is better conserved than either gag (1977 nt) or env (1713 nt). Internal deletions in coding regions were rare and limited to a loss of 1–10 nt.

Several studies have elucidated functional domains in encoded proteins for the related betaretrovirus MPMV and for SRV. Analysis of the preservation of such domains in Cer-SERV showed that many are conserved, which would predict that the capacity to encode functional proteins is likely retained in at least some Cer-SERV proviruses (Appendix A). In general, conservation values were higher for Cer-SERV-1 than for Cer-SERV-2, again suggesting an older age for Cer-SERV-2 integrations.

### 3.4. Shared SERV Proviral Integrations in Macaques

Phylogenetic analysis showed that shared Cer-SERV integrations between OWM species are rare, an observation that is in line with SERV being active during OWM speciation [4]. Although incomplete lineage sorting could partly explain a random distribution of proviruses among related species, the presence of proviruses with low LTR variation suggests that integration indeed happened until recent times, e.g., after speciation. Subsequent hybridization and introgression, however, could likely explain four Cer-SERV-1 integrations, each defective at both LTRs, at chromosomes 1, 2, 3, and 11, respectively, which are shared between the *M. mulatta* and *M. fascicularis* genomes. Both ancient and ongoing hybridization between these macaque species, with a wide active hybrid zone on the Southeast Asian mainland, has been described [45]. The rhesus and crab-eating macaques sequenced here originated from India and Indonesia, respectively, suggesting that the hybridization event that led to the shared Cer-SERV loci likely occurred in the distant past. Alternatively, as it has been proposed that both macaque species are descendants of a *M. fascicularis*-like progenitor, the alleles could have been inherited by descent [46,47].

All seven *M. fuscata* proviruses identified here, of both genotypes, can be found at the same chromosomal location in the M. mulatta genome. Such a finding is in line with the relatively short divergence time of these two closely related macaque species, which has been estimated to have occurred during the Pleistocene between 0.31–0.88 mya [48,49], suggestive of inheritance by common descent.

### 3.5. Shared Cer-SERV Proviral Integrations in Baboon and Gelada

Except for two integrations, all sites with full-length homozygous Cer-SERV proviruses in the *T. gelada* genome are empty in both *P. anubis* assemblies, suggesting they are gelada-specific or have been lost in baboons during evolution as a result of incomplete lineage sorting. The two Papionin species, *Papio* and *Theropithecus*, are estimated to have separated around 5 million years ago (mya) [45]. The first shared provirus, a full-length Cer-SERV-2 genome with integration site duplication TGGATA, is located on chromosome 17 in the gelada and in baboon 1X1155. In the *P. anubis* 15944 assembly, the chromosome 17-provirus is placed on chromosome 15. A 5′ 3575 nt partial provirus is present at the same position in *C. atys*, the sooty mangabey. In *Chlorocebus*, *Erythrocebus*, the *Macaca* spp., and *Mandrillus* spp., the integration site is empty, suggesting that the provirus integrated after the split of Papionini and Cercopithecini and after the split of Papionins from the macaque ancestor, respectively. Alternatively, it could be a heterozygous trait, or have been lost completely in some ancestral lineages. The latter scenario is more likely for *Mandrillus*, as it has been described as a sister taxon to *Cercocebus* [50]. In the gelada, the chromosome 17 provirus has the longest branch length and largest nucleotide divergence of all Cer-SERV-2 integrations (result not shown), suggesting it is the oldest full-length integration present in the genome.

A second full-length Cer-SERV-2 provirus with integration site duplication ATTTGG, present on chromosome 1 of *T. gelada* and on an unplaced scaffold of *P. anubis* 1X1155, has been reduced to a single LTR on *P. anubis* 15944 chromosome 1. Such an event may have occurred in an ancestral lineage soon after integration, with both alleles being retained in the population.

In case of the chromosome 17 Cer-SERV-2 integration, four 5′ LTRs (gelada, mangabey, and two baboon genomes) and three 3′ LTRs (gelada and two baboon genomes) can be compared to calculate integration times. Calculations show that the nt distance between the 5′ and 3′ LTRs within one of the species here is larger than between either the 5′ or 3′ LTRs between the species (Table 3), implying that the provirus resided for a longer time in the genome of a common ancestor than it did in the genome of the contemporary species. Interestingly, variation between the 5′ LTRs exceeded that between the 3′ LTRs, suggestive of a greater mutational pressure on the 5′ LTR. Using the primate ERV substitution rate r of 5 × 10^−9^ s/s/y, with regard to the large number of CpG-associated mutations being present, the integration time T_int_ of the chromosome 17 Cer-SERV-2 virus was estimated at 7.7 (7.2–8.4) mya, with subsequent divergence between baboon/gelada and mangabey occurring 3.8 (3.5–4.7) mya and between baboon and gelada 2.3 (1.3–3.7) mya. The former two ages are in line with known evolutionary time points separating the OWM species and agree with integration after the split from a macaque ancestor. Using a lower mutation rate of 2.3 × 10^−9^ s/s/y, which may be more accurate for the 3′ LTR, integration times are more than doubled (Table 3). Especially for speciation times based on 5′ LTR comparisons, the lower mutation rate results in unrealistic outcomes, such as a divergence between gelada and baboon dating to ~6.1–8 mya, with baboon speciation having occurred around ~4 mya. The slower rate likewise suggests a very distant T_int_ for the chromosome 17 Cer-SERV-2 provirus, namely around 16.8 mya, which would implicate that the integration was lost in the macaque ancestor, like it most likely was in the *Mandrillus* lineage, and in the Cercopithecines.

In contrast, using a “fast” mutation rate for 3′ LTR comparisons results in ages that are likely too recent, e.g., ~1.3 mya, as the TMRCA for baboon and gelada. Otherwise, it could point to admixture, and, indeed, analysis of Alu insertion polymorphisms has shown a long intertwined history between the species [51]. In addition, hybrids between *Papio* and *Theropithecus*, both from the wild and in captivity, are still being reported [51].

## 4. Discussion

The D-type endogenous retrovirus SERV is the most successful germ line invader of OWM genomes, both in Colobinae and in Cercopithecinae [4,7]. In the current study, the characteristics of 81 full-length Cer-SERV genomes retrieved from a number of Cercopithecinae species have been studied in detail, with regard to both viral genome features and host repressive mechanisms acting on the viral sequences. As public genome assemblies incorporate homozygous integrations only, heterozygous proviruses, which are likely to be younger, could not be studied here.

A first surprising finding was that full-length Cer-SERV proviral genomes were almost exclusively retrieved from terrestrial, savannah-living species (baboon, gelada, macaque, vervet, and red guenon), with an average of eight proviruses per genome, but not from arboreal, tropical forest-inhabiting OWM (mandrill, drill, mangabey, and De Brazza’s monkey), where only one complete provirus was found in the four species investigated. It is difficult to come up with a mechanism that would preferentially retain full-length proviral genomes in OWM sharing a habitat rather than genetics. However, it has been shown, at least for the vervet, that the majority of proviral SERV integrations are heterozygous traits [6], with lower numbers having reached fixation. Sampling a monkey from a large population in the wild may then result in few homozygous proviruses in a genome assembly. Inbreeding could increase the chance of trait fixation, such that an inbred individual may have a larger number of “visible” proviruses in its assembled genome. It is quite likely that monkeys sampled for sequencing are inbred rather than outbred, and that especially terrestrial monkeys, which are popular lab animals, have been bred in captivity, e.g., in primate centers or zoos, for quite some time. Others could descend from populations with a limited range in the wild. For instance, vervet monkeys of the Caribbean, one of which supplied the genomic sample investigated in the current study, descended from a limited number of animals that were released there a few hundred years ago [52]. Geladas, once widespread, are nowadays restricted to the Ethiopian highlands, which likewise facilitates inbreeding. Thus, the number of homozygous proviruses found in the sequenced primate genomes may not adequately represent the situation in wild, outbred monkeys.

Phylogenetic analysis confirmed the existence of two genotypes of SERV in Cercopithecinae OWM, Cer-SERV-1 and Cer-SERV-2 [3]. The two genotypes are closely related to each other, with ~88% nt homology. Both are equidistant from the Colobine variant Col-SERV, with around 72% homology over 92% of the genome, suggesting that Cer-SERV and Col-SERV followed distinct evolutionary pathways after their ancestral host species separated. Of the Cer-SERV proviruses, greater LTR divergence and a lower number of uninterrupted genes suggest that fixed germ line infections of Cer-SERV-2 are generally older than those of Cer-SERV-1. Relatively recent expansions, as illustrated by the presence of closely related clusters with short branch lengths, are seen only for Cer-SERV-1, in, e.g., *Chlorocebus*, and in one baboon genome. It is possible, however, that younger, heterozygous SERV-2 integrations exist. SERV germ line infections are limited to OWM, and have accompanied, and possibly steered, OWM speciation over time, but the evolutionary pattern of the virus is clearly complicated. Multiple processes, such as incomplete lineage sorting, hybridization between species, proviral re-activations, re-infections of the germ line, likely cross-species transmissions, and the relatively short evolutionary time elapsed since the OWM speciation process began, all muddle the picture.

No apparent selection on SERV-1 or SERV-2 genes was found when using Tajima’s Neutrality test, with all D-values being negative. However, Bhatt et al. have argued that, for RNA virus populations, Tajima’s Neutrality test performs worse than (variants of) the McDonald–Kreitman (MK) test [53]. Tajima’s test found it difficult to accommodate population expansions and different mutations occurring at the same site. A further concern with using Tajima’s D test on viral populations is that their high evolutionary rates would invalidate the test’s key assumption that each mutation occurs at a different site. However, the MK test needs an appropriate outgroup, which is difficult to define for SERV. Still, a population expansion after a selective sweep would very well describe the evolutionary scenario for successful germ-line-infecting viruses. Not all variants present in the infectious population will reach the germ line (bottleneck), after which reactivated proviruses can reinfect the germ line (population expansion), over the years accompanied by variation due to host repression and mutation (increasing diversity). SERV integrations are likely too young to have been successfully repurposed (domesticated), such that specific amino acid selection(s) would not yet be obvious. Any selection against stop codons or indels is likewise difficult to assess, since there are so many integrations, and so many ORFs are still open.

Shared Cer-SERV integrations between OWM species are rare, and even in two baboons of the same subspecies, none were found. Some, however, have been detected in three closely related macaque species, *M. mulatta*, *M. fuscata,* and *M. fascicularis*. A shared Cer-SERV-2 provirus was found on chromosome 17 in the gelada, baboon, and mangabey genomes, but not in macaques. Therefore, that provirus has likely integrated after the *Papio*/*Theropithecus*/*Cercocebus* ancestor split from the macaque ancestor around 8–10 mya. Using LTR divergence as a measurement of age suggested that the chromosome 17 provirus is one of the oldest SERV-2 integrations in the dataset, implying that most integrations occurred much more recently than 8 mya. A young age could indeed explain their absence as shared traits. These findings again point to long-term SERV activity with repeated germ line infections. Most likely, most of these integrations are still heterozygous traits, while many others may have been lost. The absence of shared proviral integrations between the two genomes of baboons apparently belonging to the same subspecies is in that respect a testament to this view.

The large number of SERV proviral integrations, with several hundreds of fixed loci having been detected in the average OWM species, profoundly alter the architecture of the OWM genomes, not only by their presence and coding capacity, but also by recombination of homologous viral sequences, and by the ability of LTRs to function as novel promoters and influence gene expression. The important role of transposable elements in reshaping mammalian development, and especially in ERV-driving lineage-specific evolution of the placenta, has been recognized [54,55]. ERVs that have lost env coding capacity can become “superspreaders”, multiplying disproportionately and taking over a considerable part of the host genome [56,57]. It is not implausible to state that SERV is a “novel” transposable element in the OWM genome for which it is yet unclear what its role and consequences for OWM evolution will be, especially as it appears to lose the env coding function.

Of course, virus invasions are counteracted by host defense mechanisms, and patterns of antiviral host pressure can be seen in endogenous viral sequences. For Cer-SERV, it was found that APOBEC3 signature patterns were almost absent, suggesting that the virus has largely been able to avoid these restriction factors during viral replication, especially APOBEC3G. The related betaretroviruses likewise actively evaded APOBEC3G but are susceptible to APOBEC3F action. In contrast, Cer-SERV proviral DNA does show abundant CpG-methylation-associated mutations, suggesting that the sequences are extensively methylated after integration in order to suppress transcription. A further repressive mechanism became apparent when PBS sequences, needed for first-strand DNA synthesis, were analyzed. Unexpectedly, Cer-SERV genotypes were found to differ in PBS use—PBS^lys3^ is used by Cer-SERV-1 and PBS^lys1,2^ by Cer-SERV-2—a variation never seen in exogenous retrovirus families. In addition, many PBS^lys1,2^ sequences were mutated. Such a flexible use of PBS sequences could have been induced by tRF repression, which is active in embryonic stem cells to suppress endogenous retrovirus transcription when, to facilitate development of the embryo, methylation patterns are erased. PBS-switching or -mutations could be an escape mechanism from such repression. It has been shown that retroviruses can tolerate mutations in the PBS without impairing replication [9]. The indications of tRF repression evasion, together with the observation that recent Cer-SERV-1 proviruses show evidence of evolving binding motifs for embryonic stem cell-specific transcription factors, such as FoxD3 and FoxJ2, it appears that SERV is further adapting from exogenous infections to a lifestyle that specializes in (re-)infection of the embryonic germ line. It is not clear whether or not such a process has a need for actual reinfection, mediated by the enveloped virus particle binding to its cellular receptor ASCT2, as the viral replication and pre-integration complex could also directly target the host genomic DNA from within the cell. ASCT2 is highly expressed on pre-implantation embryos, thus allowing receptor-mediated infection of the germ line [2,58]. However, Env expression from young, integrated proviruses should be able to block the receptor to prevent reinfection [59]. However, reinfections of the germ line are common, as indicated by the recent, species-specific integrations, and the reported abundance of heterozygous loci [6].

Over the years, it has become clear that certain retroviruses are, or have recently been, infiltrating the germ line of mice, cats, koalas, and mule deer [1,60]. During the endogenization process, those viruses retain their infectious properties. When tracing the fate of the ancient endogenous gammaretrovirus MLERV1, present in three mammalian lineages, namely bat, pangolin, and cat, Zhuo and Feschotte describe different amplification scenarios in two of the three lineages: limited proliferation from a single provirus with loss of infectious capacity soon after endogenization in the cat, and proliferation by both retrotransposition and multiple infection events in the bat [44]. They conclude that in bats, MLERV1 has repeatedly transitioned from an infectious pathogen to a genomic parasite, that is, a retrotransposon. Here, the analysis of proviral SERV genomes from monkey genomes has provided insight into an evolutionary trajectory of a monkey endogenous retrovirus to successful germ line reproduction similar to what has been described for bats. Likewise, Belshaw et al. define three mechanisms by which human ERVs with more than 200 elements proliferate: complementation *in trans* (HERV-H and ERV-9), being copied by non-HERV mechanisms (HERV-W), and retrotransposition *in cis* in germ line cells (HERV-K(HML3)) [61]. Interestingly, HERV-K(HML3) has homology to beta- and type D retroviruses [62]. It is striking to observe that yet another retrovirus is increasing its numbers through retrotransposition, or possibly reinfection, during embryogenesis. In addition, as EVEs are known to have significant effects on host evolution, it will be interesting to see if and how the numerous SERV loci influence future OWM evolution.

## 5. Conclusions

Analysis of Cer-SERV full-length proviruses present in OWM genomes suggests that this successful germ line invader is transitioning over the past millions of years from an exogenous retrovirus to an exclusive intracellular lifestyle in the early developing embryo, thereby greatly increasing the number of loci in the host genome. Indications for such a transition are:Shared, ancient proviral Cer-SERV integrations between species are rare;Most Cer-SERV proviral integrations are relatively young;Species-specific young, but not old, Cer-SERV clusters are seen in OWM genomes;Cer-SERV PBS sequences show evidence of tRF suppression, a mechanism only operational in the early embryo;Some young Cer-SERV-1 LTRs have acquired binding sites for embryo-specific transcription factors;Cer-SERV pol genes are more often uninterrupted than Cer-SERV env genes;Cer-SERV Env expression from integrated proviruses should be able to block the receptor to prevent Env-mediated reinfection of the cell. However, reinfections of the germ line are common.

It is yet unclear what the effect of the numerous SERV integrations in the OWM genome is, or will be in the future, but it is clear that SERV is an important factor influencing OWM evolution. It is also a trait that sets OWM apart from other primates.

## Figures and Tables

**Figure 1 genes-13-00119-f001:**
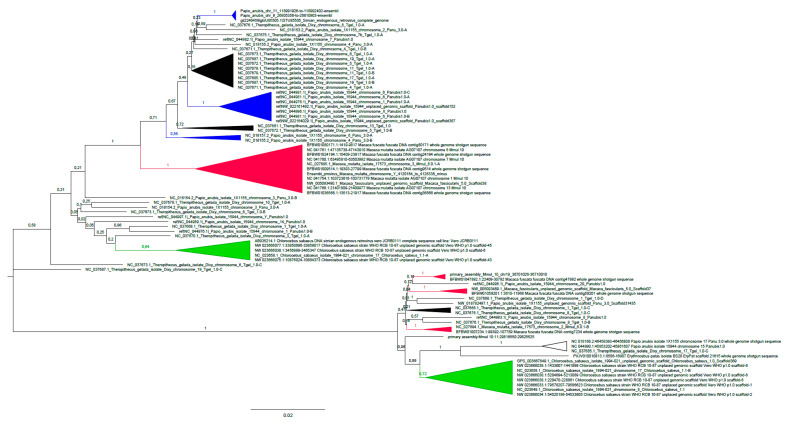
Phylogenetic analysis of 81 full-length Cer-SERV genomes. The evolutionary history was inferred by using the Maximum-Likelihood method and Hasegawa-Kishino-Yano model [12]. The tree with the highest log likelihood (−56214.44) is shown. The percentage of trees in which the associated taxa clustered together is shown next to the branches. There were a total of 8334 positions in the final dataset. Species-specific clusters have been collapsed for clarity and are indicated with colors (*Chlorocebus,* green; *Macaca,* red; *Papio,* blue; *Theropithecus,* black). The shared integration on chromosome 17 is indicated by an open triangle. Although the clustering of the *E. patas* provirus suggests it is related to the shared chromosome 17-provirus, that integration site is empty in the *E. patas* genome. An enlarged image can be found as Appendix A.

**Figure 3 genes-13-00119-f003:**
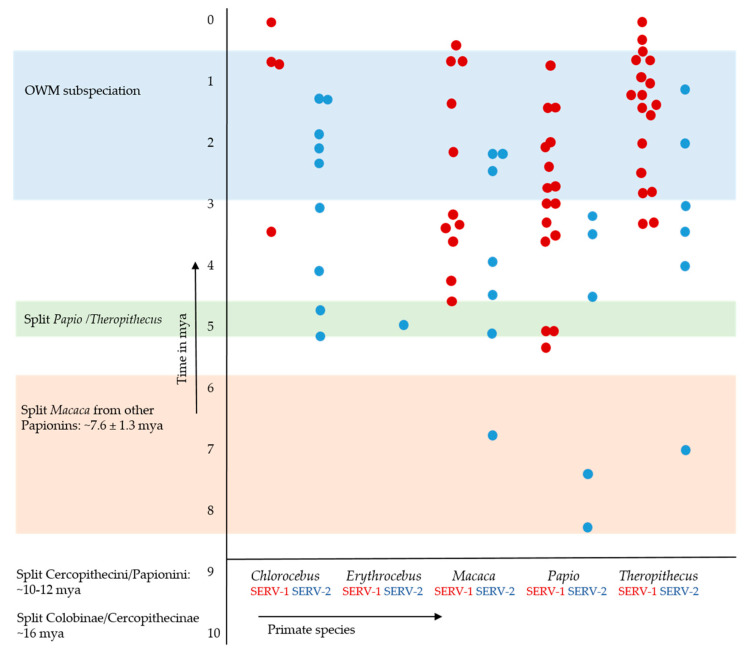
Estimated integration times of full-length, homozygous Cer-SERV proviruses in OWM genomes calculated using the “fast” mutation rate of 5 × 10^−9^ s/s/y, which appears to be the most appropriate for the dataset [43]. Using a slower rate would make the proviruses more than twice as old, with many predating accepted OWM speciation times. Estimated evolutionary events in OWM are indicated on the Y-axis (for reference, see [4]). Cer-SERV-1 is shown in red, Cer-SERV-2 in blue.

**Table 1 genes-13-00119-t001:** Characteristics of OWM genomes queried for SERV sequences, and the number of full-length viral genomes present.

OWM Species	Specimen, Gender, Origin	No. of Full-Length, HOMOZYGOUS, Cer-SERV GENOMES ^1^
*Chlorocebus aethiops* (vervet)	(a) #1994-021, male, Caribbean *C. aethiops sabeus* (b) WHO Vero cell line RCB 10–87, female *sabaeus* monkey ^2^	(a) 6 (3 SERV-1, 3 SERV-2) (b) 6 (1 SERV-1, 5 SERV-2)
*Erythrocebus patas* (red guenon)	#BS28, unknown sex, San Diego Zoo	1 (SERV-2)
*Macaca fascicularis* (crab-eating macaque)	#MacFas5, female, from Tinjil, Java, Indonesia	2 (1 SERV-1, 1 SERV-2)
*Macaca fuscata* (Japanese macaque)	#JAMA01, unknown sex, Japan	7 (4 SERV-1, 3 SERV-2)
*Macaca mulatta* (rhesus monkey)	#17573, female, Indian origin + separate Y chromosome	9 (6 SERV-1, 3 SERV-2)
*Macaca nemestrina* (pig-tailed macaque)	#M95218, female, Washington National Primate Research Center	1 (SERV-1) ^3^
*Mandrillus leucophaeus* (drill)	Isolate #KB7577, female, San Diego Zoo	1 (SERV-2) ^3^
*Papio anubis* (olive baboon)	(a) #X1155, female, Kenyan ancestry (b) #15944, male, Southwest National Primate Research Center	(a) 7 (5 SERV-1, 2 SERV-2) (b) 15 (12 SERV-1, 3 SERV-2)
*Theropithecus gelada* (gelada)	#Dixy, female, Ethiopia	26 (21 SERV-1, 5 SERV-2)

^1^ >99% of query sequence length with >90% homology to query. No proviral sequences fulfilling the criterion were found in *Cercopithecus neglectus* (De Brazza’s monkey), *Cercocebus atys* (sooty mangabey), and *Mandrillus sphinx* (mandrill), although partial proviral genomes were present. ^2^ The Vero cell line has an abnormal, hypodiploid karyotype, with 66% of cells having 2N = 58 instead of 2N = 62, and most cells having structurally altered marker chromosomes. ^3^ Sequences were of lesser quality and could not be analyzed further.

**Table 2 genes-13-00119-t002:** 5′ and 3′ LTR nucleotide distances of Cer-SERV proviruses and their calculated integration times.

Genotype/ No. of Sequences	Mean Group LTR nt Distance K ^1^	LTR nt Distance K, Range	Calculated Integration Time (Fast Rate) ^2^	Calculated Integration Time (Slow Rate) ^2^
Cer-SERV-1 *N* = 55	0.023 ± 0.014	0.000–0.053	2.3 ± 1.4 mya (range <0.3–5.3)	5.0 ± 3.0 mya (range <0.3–11.5)
Cer-SERV-2 *N* = 26	0.039 ± 0.019	0.011–0.084	3.9 ± 1.9 mya (range 2.4–18.0)	8.5 ± 4.0 mya (range 2.4–18.0)

^1^ Determined using MEGA X, with Kimura-2-parameter distances, γ-distributed, homogeneous rates among lineages, and pair-wise deletion for gaps/missing data [13]. ^2^ T = K/2r, whereby r = 5 × 10^−9^ substitutions/site/year (s/s/y) and 2.3 × 10^−9^ s/s/y, respectively [43].

**Table 3 genes-13-00119-t003:** Average age estimation based on LTR divergence of the chromosome 17 full-length Cer*-*SERV-2 integration in *Papio* (baboon 1, baboon 2), *Theropithecus* (gelada) and *Cercocebus* (mangabey).

Chromosome 17 Cer-SERV-2 LTR	K (nt Distance) ^1^	Estimated Age T of Integration ^2^	Event, with Average Estimated Age
5′/3′ LTR gelada	0.072	~7.2/~15.7 mya	Integration of chr. 17 Cer-SERV-2 provirus: ~7.7/~16.8 mya
5′/3′ LTR baboon 1 5′/3′ LTR baboon 2	0.084 0.075	~8.4/~18.3 mya ~7.5/~16.3 mya	
5′ LTR gelada/ 5′ LTR baboon 1 5′ LTR gelada/ 5′ LTR baboon 2	0.037 0.028	~3.7/~8.0 mya ~2.8/~6.1 mya	TMRCA ^3^ of gelada and baboon chr. 17 Cer-SERV-2: ~2.3/~5.6 mya
3′ LTR gelada/ 3′ LTR baboon 1 3′ LTR gelada/ 3′ LTR baboon 2	0.013 0.013	~1.3/~2.8 mya	
5′ LTR baboon 1/ 5′ LTR baboon 2	0.019	~1.9/~4.1 mya	TMRCA of two *P. anubis* chr. 17 Cer-SERV-2: <0.8/~2.2 mya
3′ LTR baboon 1/ 3′ LTR baboon 2	0.000	<0.3 mya	
5′ LTR gelada/ 5′ LTR mangabey	0.047	~4.7/~10.2 mya	TMRCA gelada/baboon/mangabey chr. 17 Cer-SERV-2: ~3.8/~8.3 mya
5′ LTR baboon 1/ 5′ LTR mangabey	0.035	~3.5/~7.6 mya	
5′ LTR baboon 2/ 5′ LTR mangabey	0.033	~3.3/~7.2 mya	

^1^ Kimura-2-parameter method, CpG sites included, complete deletion. ^2^ T = K/2r, whereby r = 5 × 10^−9^ substitutions/site/year (s/s/y) and 2.3 × 10^−9^ s/s/y, respectively [43]. ^3^ Time to Most Recent Common Ancestor.

## Data Availability

The data presented in this study are available in Appendix A.

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
