# Peer review of "Analysis of Simian Endogenous Retrovirus (SERV) Full-Length Proviruses in Old World Monkey Genomes"

_genes, 2022, doi:10.3390/genes13010119_

Round 1
Reviewer 1 Report
The author has made an extensive search for Simian Endogenous sequences in Old world monkey species genomes and analysed the full length sequences using multiple bioinformatics tools. The study has shown evidences for the transition of Cer-SERV to an intracellular propagation in the early developing embryos of Old world monkeys. Findings of this study add to the existing knowledge and will be useful to scientists working on retroviruses.
Author Response
Reviewer #1: The author has made an extensive search for Simian Endogenous sequences in Old world monkey species genomes and analysed the full length sequences using multiple bioinformatics tools. The study has shown evidences for the transition of Cer-SERV to an intracellular propagation in the early developing embryos of Old world monkeys. Findings of this study add to the existing knowledge and will be useful to scientists working on retroviruses.
Response: Reviewer #1 has no comments that require a reply. Thank you for your kind words.
Reviewer 2 Report
The article predentes by AC. Van der Kuyl, reports the screening of simian genomes for the search of simian endogenous retroviruses. Although is interesting, is very poorly presented and many interesting data is not shown.
The title does not represent what it is presented in the text.
The text is presented in a very confusing manner, where the results section is mixed with methods explanations and discussion.
Not until the very end of the results the author told us that the SERV identified are intact, wich make one wonders why then, it was necessary to analyse the possible mutations in CpG islands, and the possible APOBEC modifications of the proviruses.
The APOBEC mutation analysis was particularly confusing, since they would have happened before integration, not in the provirus.
As this reviewer was reading, the timeline appeared as a question: How old are they then? This point comes along only at the very end, and it was not really clarified.
Figure 1 where the tree cannot be read since is very packed, fonts are very small, and has a very low resolution.
A figure with the schematic of the main proviruses identified, indicating all their genetic elements and loci insertion, especially if they are in the same locus in different species, should have been included.
This article will help itself if the order in which the results are presented is changed, and if more figures and schemes are included. Also, it would help this article to properly separate methods, from results and discussion.
Author Response
Reviewer 2: The article predentes by AC. Van der Kuyl, reports the screening of simian genomes for the search of simian endogenous retroviruses. Although is interesting, is very poorly presented and many interesting data is not shown.
The title does not represent what it is presented in the text.
Response: I do not understand what is meant here? The paper analyses SERV integrations in Old World monkey genomes and presents evidence for a transition from extracellular to intracellular replication of this particular retrovirus, similar to what has been described for some betaretroviruses, so the title covers the contents of the paper. Although the other two reviewers have no comments regarding inappropriateness of the title, I have shortened it to: ‘Analysis of Simian Endogenous Retrovirus (SERV) Full Length Proviruses in Old World Monkey Genomes’.
The text is presented in a very confusing manner, where the results section is mixed with methods explanations and discussion.
Response: I agree that the paper is lengthy. Some analyses are described that cannot all be accommodated in the discussion, which would otherwise become unreadable, so a short explanation is given at the appropriate location in the Results sections. However, I have rearranged some parts (see manuscript), for instance lines 379-381 have been moved to Methods and lines 432-447 have been moved to the Discussion.
Not until the very end of the results the author told us that the SERV identified are intact, wich make one wonders why then, it was necessary to analyse the possible mutations in CpG islands, and the possible APOBEC modifications of the proviruses.
The APOBEC mutation analysis was particularly confusing, since they would have happened before integration, not in the provirus.
Response: First, I do not understand what the reviewer means with ‘not until the very end of the results the author tells us that SERV identified are intact’, as at the end of the first paragraph of the results it is clearly explained that only full length proviruses have been retrieved and are analysed. Of course, should the reviewer imply intact reading frames, then only after examining those, it was found that some viruses contain open genes for all proteins (section 3.3.2). Yet, it can never be assumed, based on sequences alone, that proviruses are intact in a replicative way, as, undetermined, functional domains or RNA structures may have been affected. And, besides having full length genomes, and/or intact reading frames, mutations have occurred, either due to APOBEC3 action before integration, or cytosine methylation after integration. In the text, I have made it more clear what happened when. It is anyway interesting to study mutation patterns in the old virus integrations, so I do not see why the reviewer believes it is unnecessary.
Lines 255-257: added: ‘Host-induced cytosine methylation of CpG dinucleotide sequences by DNA methyltransferases is a mechanism to repress RNA expression, and is commonly used to silence proviral integrations’.
Line 280: when explaining APOBEC3 action, I have added: ‘during replication, thus before integration’.
As this reviewer was reading, the timeline appeared as a question: How old are they then? This point comes along only at the very end, and it was not really clarified.
Response: Again, I do not understand what the reviewer means with this comment. The putative age of the proviruses is extensively discussed in Results section 3.3.1, Table 2 and Figure 3, which are not at the ‘very end’ of the paper?
Figure 1 where the tree cannot be read since is very packed, fonts are very small, and has a very low resolution.
Response: I am sorry. The figure does have appropriate resolution, but due to the formatting requirements by MDPI, the compression makes the font very small. I guess the problem will be solved in an online version, but I will include a larger figure 1 as a supplementary file (see legend Fig. 1).
A figure with the schematic of the main proviruses identified, indicating all their genetic elements and loci insertion, especially if they are in the same locus in different species, should have been included.
Response: I agree with the reviewer that such information would be important. However, most monkey genomes are not assembled at the chromosome level, and re-assemblies are being prepared continuously for most OWM species. This makes listing the specific localization of proviruses quite difficult and unreliable, unlike the situation for the high quality assemblies of the human genome. The problem is for instance exemplified by a shared provirus in baboons and geladas, which is localized on chromosome 17 in two assemblies (baboon 1 and gelada), and on chromosome 15 (baboon 2) in the third. I believe at present such a schematic figure as requested by reviewer 2 is difficult to generate and the information in it would be meaningless with regard to the regular updating of the genome assemblies with most currently being at the scaffold, and not at the chromosomal level.
This article will help itself if the order in which the results are presented is changed, and if more figures and schemes are included. Also, it would help this article to properly separate methods, from results and discussion.
Response: I am not sure how the order of the results should be changed? Also, the reviewer is not specific about what figures should be added. The paper already contains quite a lot of figures, tables and supplementary information, so I wonder what needs to be added? Some paragraphs, however, have been moved to more clearly separate methods, results and discussion (see revised manuscript).
Reviewer 3 Report
The manuscript describes the analysis of full length Cer-SERV genomes retrieved from the OWM genomes available. The manuscript is written in good English and in a very clear style making it easy to read and understand. The data are well presented may be with one exception of Figure 1. It would be nice to find a way to make it more readable.
I have only two minor comments:
lines 228-247 - Author shows that four proviruses that overlap with protein-coding genes are all in the reverse orientation in gelada genome. It is highly probable that proviruses oriented in sense significantly interfere with the host gene transcription (due to promoter interference and also insertion of premature RNA pol2 transcription termination signal (TTS)) thus in homozygous state they may have a detrimental effect on the protein production. Therefore, there is a selection pressure toward antisense orientation. Since the assemblies contain only homozygous genomes the real distribution of the provirus orientations in the genome including also heterozygous loci may be very different. Such bias is not seen in solo LTRs that probably do not serve as strong TTSs. The difference is not very significant due to low number of the genomes, anyway, this should be discussed.
lines 440-442 - I would appreciate showing the data separately for Cer-SERV-1 and Cer-SERV-2. If the difference between frequencies of uninterrupted env and pol genes for Cer-SERV-1 is even higher it may indicate "transposon-like" behavior of Cer-SERV-1. In such case, it may make sense to compare sequence diversity of the Cer-SERV-1 genomes with interrupted env and diversity of genomes with uninterrupted env. If the genomes with interrupted env are less diverse it would support the "transposon-like" behavior of some Cer-SERV-1 copies.
Author Response
Reviewer #3: The manuscript describes the analysis of full length Cer-SERV genomes retrieved from the OWM genomes available. The manuscript is written in good English and in a very clear style making it easy to read and understand. The data are well presented may be with one exception of Figure 1. It would be nice to find a way to make it more readable.
Response: An enlarged Fig. 1 has been added to the supplementary files.
I have only two minor comments:
lines 228-247 - Author shows that four proviruses that overlap with protein-coding genes are all in the reverse orientation in gelada genome. It is highly probable that proviruses oriented in sense significantly interfere with the host gene transcription (due to promoter interference and also insertion of premature RNA pol2 transcription termination signal (TTS)) thus in homozygous state they may have a detrimental effect on the protein production. Therefore, there is a selection pressure toward antisense orientation. Since the assemblies contain only homozygous genomes the real distribution of the provirus orientations in the genome including also heterozygous loci may be very different. Such bias is not seen in solo LTRs that probably do not serve as strong TTSs. The difference is not very significant due to low number of the genomes, anyway, this should be discussed.
Response: Thank you for this remark. Added after line 233: , suggestive of a negative selection of forward orientated integrations, possibly because of the introduction of RNA polymerase II transcription termination sites (TTS) provided by the LTR (Conley & Jordan, 2012). Such transposable element-associated TTS are known to cause premature, aberrant termination of host gene transcription. I have also added to the next sentence, on a provirus in the forward orientation next to a tRNA gene exon: ‘which is probably less of a problem here, since tRNA genes are transcribed by RNA polymerase III’.
lines 440-442 - I would appreciate showing the data separately for Cer-SERV-1 and Cer-SERV-2. If the difference between frequencies of uninterrupted env and pol genes for Cer-SERV-1 is even higher it may indicate "transposon-like" behavior of Cer-SERV-1. In such case, it may make sense to compare sequence diversity of the Cer-SERV-1 genomes with interrupted env and diversity of genomes with uninterrupted env. If the genomes with interrupted env are less diverse it would support the "transposon-like" behavior of some Cer-SERV-1 copies.
Response: Lines 440-442 (revised lines 452-454): the percentages for the two Cer-SERV types replace the combined data: ‘It is of interest to note that, compared with gag (Cer-SERV-1: 55%; Cer-SERV-2: 23%) and pol (Cer-SERV-1: 71%; Cer-SERV-2: 39%), a lower percentage of Cer-SERV-1 proviruses has an uninterrupted env gene (Cer-SERV-1: 40%; Cer-SERV-2: 27%),’
To analyse mean diversity in Cer-SERV-1 genomes with and without intact env gene, as suggested by the reviewer, could be meaningful. I have first analysed nucleotide variation within T. gelada, which is probably a better estimate than comparisons of SERV between species. In the gelada genome, homozygous Cer-SERV-1 integrations with an intact env did not differ in genome diversity (within mean group nt distance = 0.029) compared to those with an interrupted env (within mean group nt distance = 0.030). Similar results were obtained when using the complete Cer-SERV-1 dataset (within group nt distance with env intact = 0.037 for whole genome comparison; with env interrupted mean within group distance = 0.038). However, numbers are relatively small, heterozygous insertions are missing, and not all sequencing information is complete reliable for the various genome assemblies. Cer-SERV-1 may be in the very early phase of transposon-like behaviour, and such recent developments are not be adequately captured by the current investigation. It is not unlikely to assume, taking into account the relatively recent integration activities of SERV, that any expansion of a SERV genome with a non-functional env gene within a species would still be in a heterozygous state.
Round 2
Reviewer 2 Report
x